# The redox-responsive transcriptional regulator Rex represses fermentative metabolism and is required for *Listeria monocytogenes* pathogenesis

Cortney R. Halsey⬤, Rochelle C. Glover⬤, Maureen K. Thomason, Michelle L. Reniere⬤*

Department of Microbiology, University of Washington School of Medicine, Seattle, Washington, United States of America

* reniere@uw.edu

**Data Availability Statement:** All relevant data are within the manuscript and its Supporting Information files.

## Abstract

The Gram-positive bacterium *Listeria monocytogenes* is the causative agent of the foodborne disease listeriosis, one of the deadliest bacterial infections known. In order to cause disease, *L. monocytogenes* must properly coordinate its metabolic and virulence programs in response to rapidly changing environments within the host. However, the mechanisms by which *L. monocytogenes* senses and adapts to the many stressors encountered as it transits through the gastrointestinal (GI) tract and disseminates to peripheral organs are not well understood. In this study, we investigated the role of the redox-responsive transcriptional regulator Rex in *L. monocytogenes* growth and pathogenesis. Rex is a conserved canonical transcriptional repressor that monitors the intracellular redox state of the cell by sensing the ratio of reduced and oxidized nicotinamide adenine dinucleotides (NADH and NAD$^+$, respectively). Here, we demonstrated that *L. monocytogenes* Rex represses fermentative metabolism and is therefore required for optimal growth in the presence of oxygen. We also show that *in vitro*, Rex represses the production of virulence factors required for survival and invasion of the GI tract, as a strain lacking *rex* was more resistant to acidified bile and invaded host cells better than wild type. Consistent with these results, Rex was dispensable for colonizing the GI tract and disseminating to peripheral organs in an oral listeriosis model of infection. However, Rex-dependent regulation was required for colonizing the spleen and liver, and *L. monocytogenes* lacking the Rex repressor were nearly sterilized from the gallbladder. Taken together, these results demonstrated that Rex functions as a repressor of fermentative metabolism and suggests a role for Rex-dependent regulation in *L. monocytogenes* pathogenesis. Importantly, the gallbladder is the bacterial reservoir during listeriosis, and our data suggest redox sensing and Rex-dependent regulation are necessary for bacterial survival and replication in this organ.

**Funding:** Research in the Reniere Lab is funded by NIH grant R01 AI132356 (MLR). C.R.H. is funded by NIH grant T32AI055396. R.C.G. is funded by NIH grant 5T32AI055396. This work used the Genomics and Bioinformatics Shared Resources at Fred Hutchinson Cancer Research Center which is partially funded from Cancer Center grant NCI 5P30CA015704-43. The funders had no role in study design, data collection and analysis, decision to publish, or preparation of the manuscript.

**Competing interests:** The authors have declared that no competing interests exist.

## Author summary

Listeriosis is a foodborne illness caused by *Listeria monocytogenes* and is one of the deadliest bacterial infections known, with a mortality rate of up to 30%. Following ingestion of contaminated food, *L. monocytogenes* disseminates from the gastrointestinal (GI) tract to peripheral organs, including the spleen, liver, and gallbladder. In this work, we investigated the role of the redox-responsive regulator Rex in *L. monocytogenes* growth and pathogenesis. We demonstrated that alleviation of Rex repression coordinates expression of genes necessary in the GI tract during infection, including fermentative metabolism, bile resistance, and invasion of host cells. Accordingly, Rex was dispensable for colonizing the GI tract of mice during an oral listeriosis infection. Interestingly, Rex-dependent regulation was required for bacterial replication in the spleen, liver, and gallbladder. Taken together, our results demonstrate that Rex-mediated redox sensing and transcriptional regulation are important for *L. monocytogenes* metabolic adaptation and virulence.

## Introduction

To successfully colonize different niches, bacteria must be able to rapidly sense and respond to environmental changes. The Gram-positive bacterium *Listeria monocytogenes* is an excellent example of this adaptability. As a saprophyte and intracellular pathogen, *L. monocytogenes* coordinates its metabolic and virulence programs to transition from life in nature to the mammalian host where it causes the foodborne disease listeriosis. Following ingestion of contaminated foods by the host, *L. monocytogenes* contends with acid stress in the stomach and acidic bile in the small intestine before descending to the cecum where it traverses the intestinal barrier [1]. Traveling via the lymph or blood, *L. monocytogenes* disseminates to the spleen and liver where it replicates intracellularly. The intracellular lifecycle requires *L. monocytogenes* to quickly escape the oxidizing vacuolar compartment to replicate in the highly reducing environment of the cytosol and then spread cell-to-cell [2–5]. From the liver, the bacteria enter the gallbladder and replicate extracellularly to very high densities and then reseed the intestinal tract upon bile secretion [6–8]. Bile itself is antimicrobial, acting as a detergent that disrupts bacterial membranes and denatures proteins [9]. Although *L. monocytogenes* virulence determinants have been investigated for decades, the vast majority of studies injected the bacteria intravenously rather than infecting mice through the natural foodborne route [10, 11]. Therefore, the mechanisms by which *L. monocytogenes* senses and adapts to the many stressors of the host environment during oral infection are not well understood.

Given its ability to replicate in diverse environmental niches, it is critical for *L. monocytogenes* to appropriately modify its metabolism in response to the changing extracellular surroundings. Reduced and oxidized nicotinamide adenine dinucleotides (NADH and $NAD^+$, respectively) play important roles in many biological processes and are therefore key molecules for sensing the intracellular redox state [12]. For example, during aerobic respiration the $NADH:NAD^+$ ratio is kept low as NADH is oxidized to $NAD^+$ by the electron transport chain (ETC). In hypoxic environments or when the ETC is inhibited, NADH levels become elevated and $NAD^+$ is no longer available to fuel carbon oxidation for growth. Therefore, the $NADH:NAD^+$ ratio is the primary indicator of the metabolic state of a cell.

The transcriptional repressor Rex monitors the intracellular redox state of the cell by directly sensing the $NADH:NAD^+$ ratio and repressing target genes when this ratio is low [13, 14]. An increase in relative NADH abundance following reduced respiration results in Rex dissociating from DNA and derepression of target genes [15–18]. Rex is widely conserved across

Gram-positive bacteria and while there is considerable variability in the identity of Rex-dependent genes among organisms, Rex generally functions to regulate metabolic pathways involved in NAD$^+$-regeneration, such as fermentation [13, 14].

*L. monocytogenes* encodes a Rex protein that shares 65% and 56% identity with homologues in *Bacillus subtilis* and *Staphylococcus aureus* [19]. We hypothesized *L. monocytogenes* Rex may be important during infection to sense the changing environment and regulate metabolic pathways accordingly. In this study, we identified Rex-dependent transcriptional changes in *L. monocytogenes* and demonstrated a role for Rex regulation during oral listeriosis.

## Results

### Transcriptomics identifies Rex-regulated genes

To investigate the role of Rex in *L. monocytogenes*, we generated a Δ*rex* mutant via allelic exchange and analyzed the Rex-dependent transcriptome under standard growth conditions. RNA sequencing (RNA-seq) was performed on RNA harvested from mid-log and stationary phase cultures of wild type (wt) and Δ*rex* strains grown aerobically in brain heart infusion (BHI) broth (Tables A-D in S1 Text). We did not observe dramatic growth phase-dependent differences in Rex-dependent regulation so here we focus on the stationary phase results for simplicity. In the Δ*rex* mutant, 196 transcripts were significantly increased in abundance at least two-fold ($p < 0.01$), indicating these genes are repressed by Rex during aerobic growth (Table 1 and Table A in S1 Text). Some of the most dramatically increased transcripts were involved in fermentative metabolism, including those encoding alcohol dehydrogenase (*lap*), pyruvate formate lyase (*pflA* and *pflBC)*, and lactate dehydrogenase (*ldhA*) (Table 1). Unexpectedly, transcripts encoding virulence factors involved in bile resistance (*bsh*, bile salt hydrolase) and host cell invasion (*inlAB*, internalin A and B) were in greater abundance in the Δ*rex* mutant, indicating Rex-dependent regulation may impact virulence. The RNA-seq results were validated via quantitative RT-PCR (qPCR) by measuring the expression of 5 genes in wt and Δ*rex* during stationary phase aerobic growth. We also measured gene expression in the

**Table 1. Rex-repressed genes-of-interest.**

| 10403S | EGD-e | Gene | Function | Fold change in Δ*rex* |
|---|---|---|---|---|
| LMRG_01332 | lmo1634 | *lap* | bifunctional acetaldehyde-CoA/alcohol dehydrogenase | 342.30 |
| LMRG_00859 | lmo1407 | *pflC* | pyruvate formate-lyase 1-activating enzyme | 88.30 |
| LMRG_00858 | lmo1406 | *pflB* | formate acetyltransferase | 59.84 |
| LMRG_00046 | lmo0355 | *frdA* | fumarate reductase flavoprotein subunit | 85.24 |
| LMRG_01064 | lmo1917 | *pflA* | formate acetyltransferase | 77.91 |
| LMRG_01979 | lmo2717 | *cydB* | cytochrome d ubiquinol oxidase subunit II | 19.05 |
| LMRG_01980 | lmo2716 | *cydC* | ABC transporter | 17.62 |
| LMRG_01981 | lmo2715 | *cydD* | ABC transporter | 15.76 |
| LMRG_01978 | lmo2718 | *cydA* | cytochrome bd-I oxidase subunit I | 13.99 |
| LMRG_01659 | lmo2173 | - | sigma-54-dependent transcriptional regulator | 16.62 |
| LMRG_00127 | lmo0434 | *inlB* | internalin B | 10.97 |
| LMRG_00126 | lmo0433 | *inlA* | internalin A | 10.26 |
| LMRG_01801 | lmo2447 | - | Rgg/GadR/MutR family transcriptional regulator | 10.76 |
| LMRG_02012 | lmo0912 | - | formate transporter | 10.71 |
| LMRG_01217 | lmo2067 | *bsh* | bile acid hydrolase | 5.57 |
| LMRG_02632 | lmo0210 | *ldhA* | lactate dehydrogenase | 3.30 |

Highlighted genes are predicted to be in an operon [20].

*rex* complemented strain, in which *rex* was expressed from its native promoter at an ectopic site in the chromosome (Δ*rex* p-*rex*). In the Δ*rex* mutant, all 5 transcripts were increased in abundance compared to the wt and Δ*rex* p-*rex* strains (S1 Fig), consistent with the RNA-seq results.

*In silico* promoter analysis of genes in the 10403S genome exhibiting Rex-dependent regulation was performed to determine potential Rex binding sites using the *Bacillus subtilis* Rex consensus sequence [13]. Allowing up to 3-mismatches with the *B. subtilis* consensus sequence, we identified potential Rex binding sites in the promoter regions of 48 genes and/or operons repressed by Rex (Table E in S1 Text). Specifically, we identified putative Rex binding sites upstream of *lap*, *pflBC*, and *pflA*, indicating Rex likely represses fermentative metabolism directly. Rex binding sites were also predicted upstream of *bsh* and *inlAB*, further suggesting direct involvement of Rex in virulence gene regulation.

In the absence of *rex*, 110 transcripts were less abundant during aerobic growth, indicating the presence of Rex is required to fully activate these genes (Table C in S1 Text). As Rex is a canonical transcriptional repressor, we hypothesize these changes are due to indirect effects. Indeed, promoter analysis did not identify any putative Rex binding sites in the promoters of genes activated in the presence of Rex, suggesting these changes are likely due to indirect Rex-dependent regulation.

## Fermentative metabolism is repressed by Rex

Transcriptional analysis indicated that Rex-mediated repression functions to down-regulate fermentative metabolism during aerobic growth (Fig 1A). To verify the role of Rex in regulating metabolism, we first assessed growth of the wt and Δ*rex* strains during both aerobic and anaerobic growth. A small, but significant, growth defect was observed for *rex*-deficient *L. monocytogenes*, beginning 4 hours post-inoculation into aerobic shaking flasks (Fig 1B). This defect was not due to a change in glucose uptake, as wt and Δ*rex* consumed glucose similarly (Fig 1C). In contrast, the Δ*rex* strain exhibited no growth defect when grown anaerobically (S2A Fig), demonstrating Rex-dependent repression is alleviated during wt anaerobic growth.

To clarify the effect of Rex regulation on *L. monocytogenes* aerobic growth, extracellular metabolites were quantified 4 hours post-inoculation. The Δ*rex* mutant secreted approximately 90% more lactate and ~55% more formate than wt and the complemented strain (Fig 1D and 1E). This was accompanied by a concomitant decrease in the primary aerobic by-product acetate (Fig 1F). These metabolite changes confirmed the transcriptional analysis showing increased transcript abundance of *ldhA*, *pflA*, and *pflBC* in the absence of *rex* and demonstrated that carbon-flux is being directed primarily towards lactate fermentation (Fig 1A). When grown anaerobically, both the wt and Δ*rex* strains produced more lactate and formate, demonstrating a switch to fermentative metabolism in the absence of oxygen (Fig 1D and 1E). We were unable to determine if the increased expression of *lap* in Δ*rex* impacted ethanol production under aerobic conditions, as this volatile metabolite evaporates in a shaking flask and could not be reliably quantified. However, ethanol was detected when bacteria were grown anaerobically in a sealed tube. During mid-log growth, the wt and Δ*rex* strains produced similar amounts of ethanol (S2F Fig). However, following glucose depletion and entry into stationary phase at 6 hours (S2G Fig), the Δ*rex* mutant made significantly more ethanol than wt (S2H Fig). These results were not unexpected, as *lap* expression was increased in the Δ*rex* mutant following 7 hours of anaerobic growth (S1B Fig), indicating that Rex functions to repress *lap* transcription when bacteria enter stationary phase.

Results from the extracellular metabolite analysis led us to hypothesize that the aerobic growth defect exhibited by the Δ*rex* strain is a result of aberrant carbon flux through

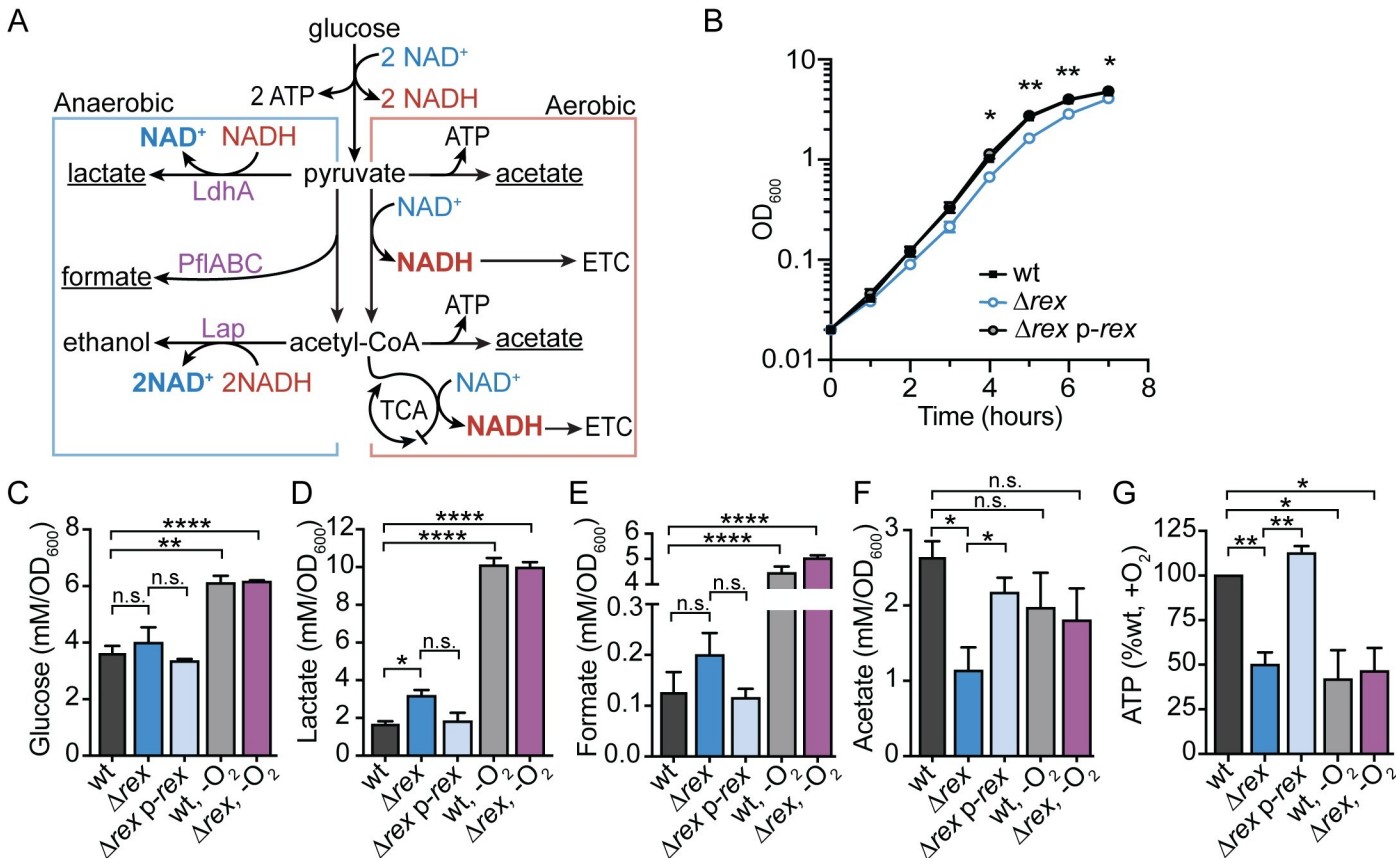

**Fig 1. Fermentative metabolism is repressed by Rex during aerobic growth.** A. Model of aerobic and anaerobic central metabolic pathways in *L. monocytogenes*. Enzymes encoded by genes repressed by Rex are denoted in purple text. Underlined metabolic end-products were those differentially produced by Δ*rex* compared to wt during aerobic growth. LdhA, lactate dehydrogenase; PflABC, pyruvate formate lyase; Lap, alcohol dehydrogenase; ATP, adenosine triphosphate; NAD(H), nicotinamide adenine dinucleotide; ETC, electron transport chain; TCA, tricarboxylic acid cycle. B. Aerobic growth of wt, Δ*rex*, and the complemented strain (Δ*rex* p-*rex*) measured by optical density ($OD_{600}$). C-F. Glucose and extracellular metabolites were quantified 4 hours post-inoculation in aerobic (black, blue, and light blue bars) and anaerobic (grey and purple bars) cultures. Concentrations of glucose (C), lactate (D), formate (E), and acetate (F) were determined and normalized to $OD_{600}$. G. Relative intracellular ATP concentration was measured at 4 hours during aerobic (black, blue, and light blue bars) and anaerobic (grey and purple bars) growth. Data are the means and standard error of the mean (SEM) of three independent experiments in all panels with the exception of panel G. Here, anaerobic samples are the means and SEMs of two independent experiments. *p* values were calculated using a heteroscedastic Student's *t* test. * $p < 0.05$; ** $p < 0.01$; **** $p < 0.0001$; n.s. $p > 0.05$. In all panels, no significant difference was found between wt and Δ*rex* p-*rex*. In C-G, no significant difference was found between the wt and Δ*rex* strains grown in anaerobic conditions.

fermentation. This would result in decreased intracellular ATP stores compared to wt aerobic growth which generates ATP through oxidative phosphorylation and the ETC (Fig 1A). To test this hypothesis, we measured ATP concentrations 4 hours post-inoculation when the growth defect of Δ*rex* becomes apparent. Indeed, the *rex*-deficient strain had 50% the amount of ATP compared to wt and the complemented strain in aerobic conditions (Fig 1G). Similarly, strains grown anaerobically had ~50% the amount of ATP compared to wt growing aerobically, further demonstrating the Δ*rex* strain is undergoing fermentation during aerobic growth (Fig 1G). As previously stated, glucose consumption and extracellular metabolite profiles were similar between the wt and Δ*rex* strains when incubated anaerobically, demonstrating that Rex-mediated repression is normally alleviated in this growth environment (Figs 1C–1G and S2). Taken together, these data indicate that *L. monocytogenes* Rex functions to repress fermentative metabolism in the presence of oxygen and a strain lacking *rex* is impaired for aerobic growth as a result of altered carbon-flux and decreased ATP production.

### *L. monocytogenes Δrex* is more resistant to acidified bile *in vitro*

In addition to metabolic genes, the transcriptional profile revealed that Rex represses virulence determinants necessary in the host gastrointestinal (GI) tract, including bile salt hydrolase (*bsh*) and internalins A and B (*inlAB*). Bsh detoxifies bile, which is encountered during transit through the GI tract and colonization of the gallbladder [21–23]. To investigate the role of Rex in bile resistance, we first generated *L. monocytogenes* strains lacking *bsh* or both *rex* and *bsh*, and assessed their survival following a 24-hour exposure to porcine bile in BHI. *L. monocytogenes* colonizing the gallbladder would be exposed to bile at neutral pH [9], which we found to have no effect on the survival of any bacterial strain tested (Fig 2). These results are consistent with published reports demonstrating that *L. monocytogenes bsh* is not required for survival in neutral bile [22, 23]. In contrast, *L. monocytogenes* in the GI tract encounters acidified bile following its release from the gallbladder into the low pH environment of the duodenum [9, 24]. Acidified bile was highly bactericidal, resulting in a 2.5-log reduction in wt survival 24 hours post-inoculation (Fig 2). This killing was dependent on bile, as all strains grew equally well in acidified BHI lacking bile (S3 Fig). *L. monocytogenes* lacking *bsh* was even more sensitive to acidified bile, exhibiting a 4-log decrease in survival (Fig 2). As predicted, the Δ*rex* mutant was more resistant to the toxic effects of acidified bile, exhibiting only a ~9-fold reduction in CFU. Trans-complementation of *rex* returned the susceptibility to similar levels as wt (Δ*rex* p-*rex*). Moreover, the Δ*rex*Δ*bsh* double mutant displayed similar susceptibility as the Δ*bsh* mutant (Fig 2), indicating the increased survival of the Δ*rex* mutant is dependent upon *bsh* expression. These results demonstrated that *L. monocytogenes* lacking the Rex repressor are more resistant to acidified bile due to increased *bsh* expression.

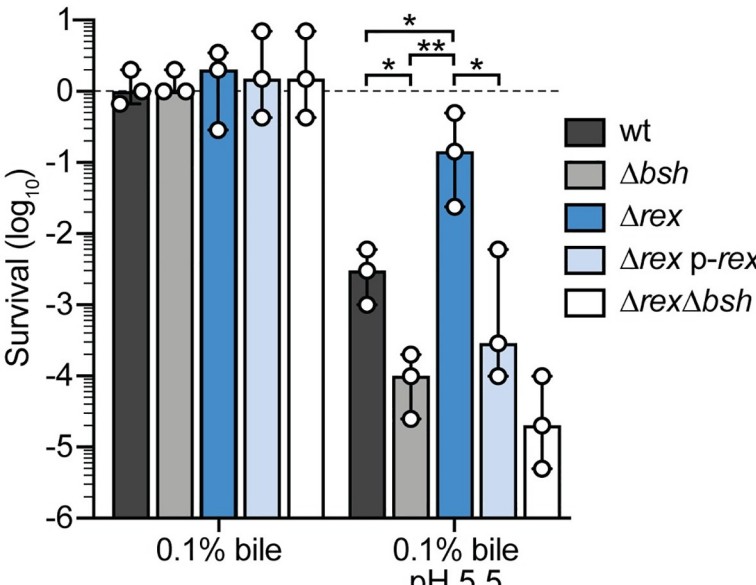

**Fig 2. Alleviation of Rex repression promotes bacterial survival in acidified bile.** Survival of wt (black), Δ*bsh* (grey), Δ*rex* (blue), Δ*rex* p-*rex* (light blue) and Δ*rex*Δ*bsh* (white) normalized to the initial inoculum (dashed line = 1) and expressed as log-transformed CFU per mL of culture. Strains were evaluated for survival 24 hours post-inoculation in BHI supplemented with bile or acidified BHI supplemented with bile under aerobic conditions. Data are the means and ranges of three independent experiments. *p* values were calculated using a heteroscedastic Student's *t* test. * $p < 0.05$; ** $p < 0.01$; n.s. $p > 0.05$. No significant difference was found between the Δ*bsh* and Δ*rex*Δ*bsh* strains.

## The role of Rex in the intracellular lifecycle of *L. monocytogenes*

The intracellular lifecycle of *L. monocytogenes* begins with entry into host cells by phagocytosis or bacterial-mediated invasion, followed by replication within the cytosol, and cell-to-cell spread via actin polymerization [2]. RNA-seq revealed that transcripts encoding InlA and InlB were increased in the Δ*rex* mutant, leading to the hypothesis that Rex regulates invasion of non-phagocytic cells via receptor-mediated endocytosis. Specifically, InlA and InlB mediate invasion of epithelial cells and hepatocytes by engaging the host receptors E-cadherin and Met, respectively [5, 25]. To investigate the effects of increased *inlAB* transcription in the Δ*rex* mutant, we measured bacterial invasion of Caco-2 human intestinal epithelial cells and Huh7 human hepatocytes. To measure invasion, cells were treated with gentamicin 1 hour post-infection and intracellular bacteria were enumerated 2 hours post-infection. We observed a significant 41% increase in invasion of Caco-2 cells by the Δ*rex* strain, which could be reduced back to wt levels with ectopic expression of *rex* (Fig 3A). This increase in *Δrex* was completely dependent on *inlAB* expression, as invasion was not significantly different between the Δ*inlAB* and Δ*rex*Δ*inlAB* strains. Similarly, Δ*rex* exhibited increased invasion of Huh7 cells in an InlAB-dependent manner (Fig 3B). Together, these results demonstrated that increased *inlAB* transcription in *L. monocytogenes Δrex* results in increased invasion of human intestinal epithelial cells and human hepatocytes *in vitro*.

After invading host cells via receptor-mediated endocytosis or phagocytosis, *L. monocytogenes* replicates intracellularly and spreads cell-to-cell using actin-based motility [2–5]. To investigate the role of Rex regulation in these facets of pathogenesis, we first measured intracellular growth in several relevant cell types. We found that the Δ*rex* strain replicated intracellularly at the same rate as wt in activated bone marrow-derived macrophages (BMMs), Huh7 human hepatocytes, and TIB73 murine hepatocytes (Fig 4A–4C). These results suggested that

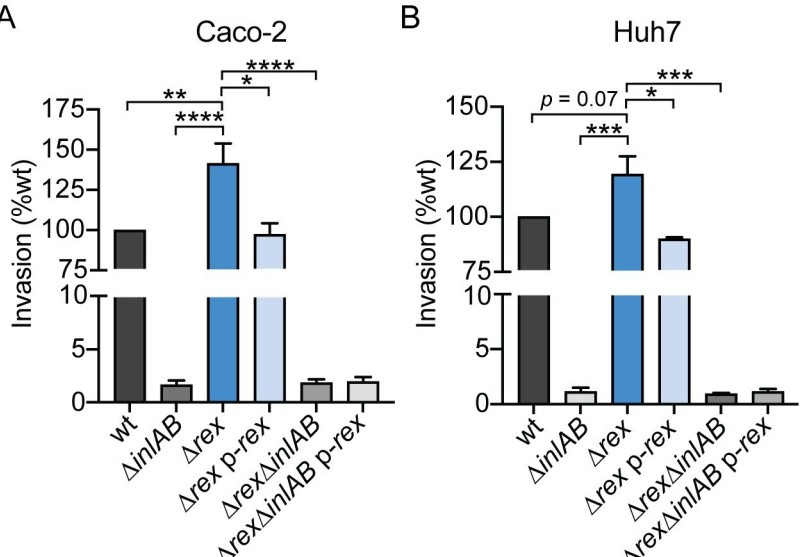

**Fig 3. Alleviation of Rex repression promotes bacterial invasion of Caco-2 epithelial cells and Huh7 hepatocytes.**
The ability of *L. monocytogenes* strains to invade Caco-2 epithelial cells (A) and Huh7 hepatocytes (B) was evaluated. Bacterial invasion was measured 2 hours post-infection and 1 hour after adding gentamicin to kill extracellular bacteria. Invasion was normalized to the initial inocula and is reported as a percentage of wt. In panel A, data are the means and SEMs of 5 independent experiments performed in duplicate. In panel B, data are the means and SEMs of 3 independent experiments performed in duplicate. *p* values were calculated using a heteroscedastic Student's *t* test. * $p < 0.05$; ** $p < 0.01$; *** $p < 0.001$. No statistical significance was found between strains lacking *inlAB*.

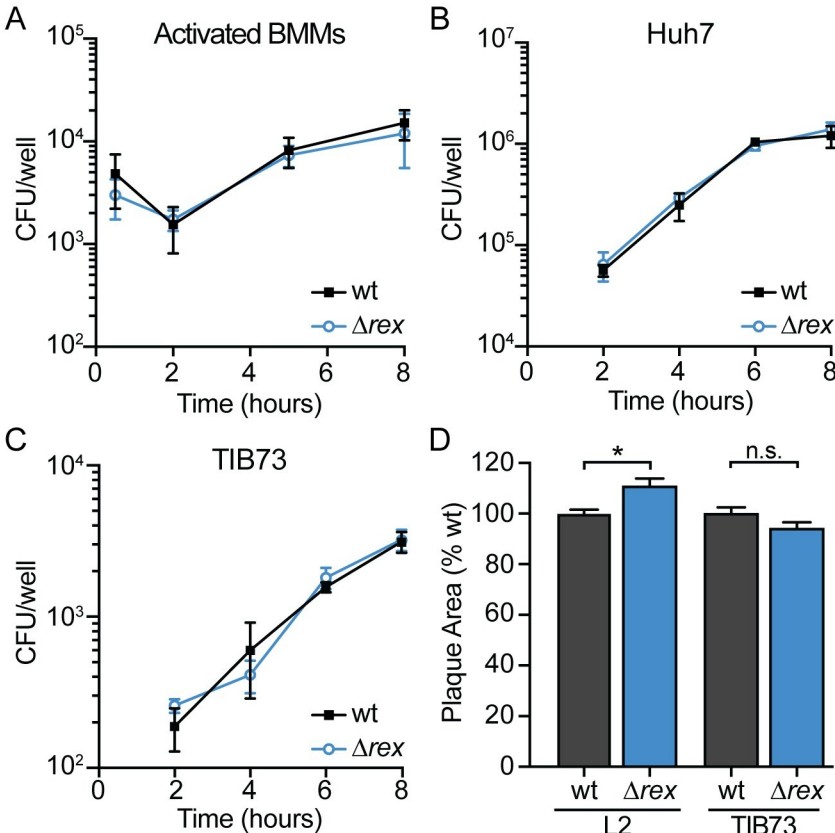

**Fig 4. Intracellular growth and cell-to-cell spread are not impaired in *Δrex*.** A-C. Intracellular growth kinetics of wt and *Δrex* in IFNγ-activated BMMs (A), Huh7 cells (B), and TIB73 cells (C). D. Plaque area in L2 fibroblasts and TIB73 hepatocytes, measured as a percentage of wt. Data in panels A and D are the means and SEMs of three independent experiments. Data in panels B and C are the means and SEMs of two independent experiments performed in duplicate. In all panels, *p* values were calculated using a heteroscedastic Student's *t* test. * $p < 0.05$; n.s. $p > 0.05$.

Rex-mediated repression is not required for intracellular growth and therefore, deleting the Rex repressor had no effect on intracellular replication.

Next, cell-to-cell spread was evaluated via plaque assays in which a monolayer of cells is infected and both intracellular growth and intercellular spread are measured over 3 days [26]. The *Δrex* mutant formed plaques ~10% larger than those formed by wt *L. monocytogenes* in L2 murine fibroblasts and formed plaques ~5% smaller in TIB73 hepatocytes (Fig 4D), indicating that Rex regulation is not required for cell-to-cell spread. Taken together, these data demonstrated that *L. monocytogenes* lacking the Rex repressor has an advantage at early stages of infection, as *Δrex* displayed increased invasion of host cells. However, Rex is dispensable for intracellular growth and intercellular spread in all cell types analyzed.

## Rex is required for virulence in a murine oral model of infection

We hypothesized that during oral infection of *L. monocytogenes*, Rex-mediated repression is alleviated due to the hypoxic environment of the GI tract. Derepression of Rex target genes would not only up-regulate fermentative metabolism for energy production in this environment but would also increase transcription of virulence factors required for successful infection of the GI tract. To test the ability of *Δrex* to survive in the murine GI tract, 6- to 8-week old female BALB/c mice were orally infected with $10^8$ CFU of wt, *Δrex*, or *Δrex* p-*rex* and

housed in cages with elevated wire bottoms to limit reinoculation by coprophagy [27, 28]. Prior to infection, mice were treated with streptomycin for 48 hours to increase susceptibility to oral *L. monocytogenes* infection [8, 27, 28]. Changes in body weight were recorded throughout the infection as a global measurement of disease severity [8]. Mice infected with wt and Δ*rex* p-*rex L. monocytogenes* lost ~12% of their initial body weight throughout the 4 day infection (Fig 5A). In contrast, mice infected with Δ*rex* lost only ~3% of their initial weight 3 days following infection and returned to their initial weight by 4 days post-infection (Fig 5A). These results indicated mice infected with Δ*rex* experienced less severe disease than mice infected with either wt or the complemented strain following oral infection.

To determine bacterial burden, organs and feces were harvested, homogenized, and plated. Specifically within the GI tract, we analyzed the small intestinal tissue, intestinal contents, cecum, and feces. Similar bacterial loads were observed between all strains in the GI tract and feces throughout the infection, indicating that Rex-mediated transcriptional repression is dispensable for the GI phase of infection, as predicted (Fig 5B–5E). Rex was also not required for dissemination from the GI tract to internal organs, as evidenced by the similar bacterial burdens between all strains in the spleen, liver, and gallbladder 1 day post-infection (Fig 5F–5H). However, by day 4 of the infection, we observed a significant attenuation of Δ*rex* compared to wt. Bacterial burdens in the spleens and livers of mice infected with Δ*rex* were decreased by approximately 1-log compared to the wt strain (Fig 5F and 5G). The most dramatic attenuation was observed in the gallbladder, with Δ*rex* decreased approximately 5-logs compared to wt 4 days post-infection (Fig 5H). Importantly, virulence could be restored back to wt levels in all three organs when mice were infected with the *rex* complemented strain (Fig 5F–5H). We also assessed bacterial burdens in mice infected with either the wt or Δ*rex* strains over the course of a 4-day infection in two additional independent experiments and found similar results in all organs (S4 and S5 Figs). Taken together, these results confirmed our hypothesis that Rex-dependent repression is not required for colonization and invasion of the GI tract during oral infection, as a Δ*rex* mutant was able to colonize and disseminate similar to wt. These results suggest that Rex is normally derepressed in this anaerobic environment. In addition, the infection studies revealed that Δ*rex* is able to disseminate to internal organs in the early stages of infection. However, Rex-dependent regulation was required for replication in the spleen, liver, and gallbladder after oral infection.

## Discussion

In this study, we investigated the role of the redox-responsive transcriptional regulator Rex in *L. monocytogenes*. Transcriptional and *in silico* promoter analyses identified dozens of genes likely to be directly repressed by Rex *in vitro*. We demonstrated that derepression of Rex target genes induces fermentative metabolism, resulting in decreased ATP production and impaired aerobic growth of *L. monocytogenes* lacking *rex*. We also present evidence that the presence of Rex impacts virulence factor production *in vitro*. These studies revealed that Δ*rex* is more resistant to acidified bile in a Bsh-dependent manner and that over-expression of *inlAB* in the Δ*rex* mutant leads to increased invasion of host cells. *In vivo* experiments demonstrated that Rex is dispensable for colonizing the GI tract and disseminating to peripheral organs in an oral listeriosis model of infection. However, Rex was required for colonization of the spleen, liver, and gallbladder. This *in vivo* attenuation was not a result of impaired intracellular replication or cell-to-cell spread, as the Δ*rex* mutant performed similar to wt in cell culture assays of infection. Taken together, our results indicate an important role for redox sensing and Rex-mediated transcriptional repression during *L. monocytogenes* infection.

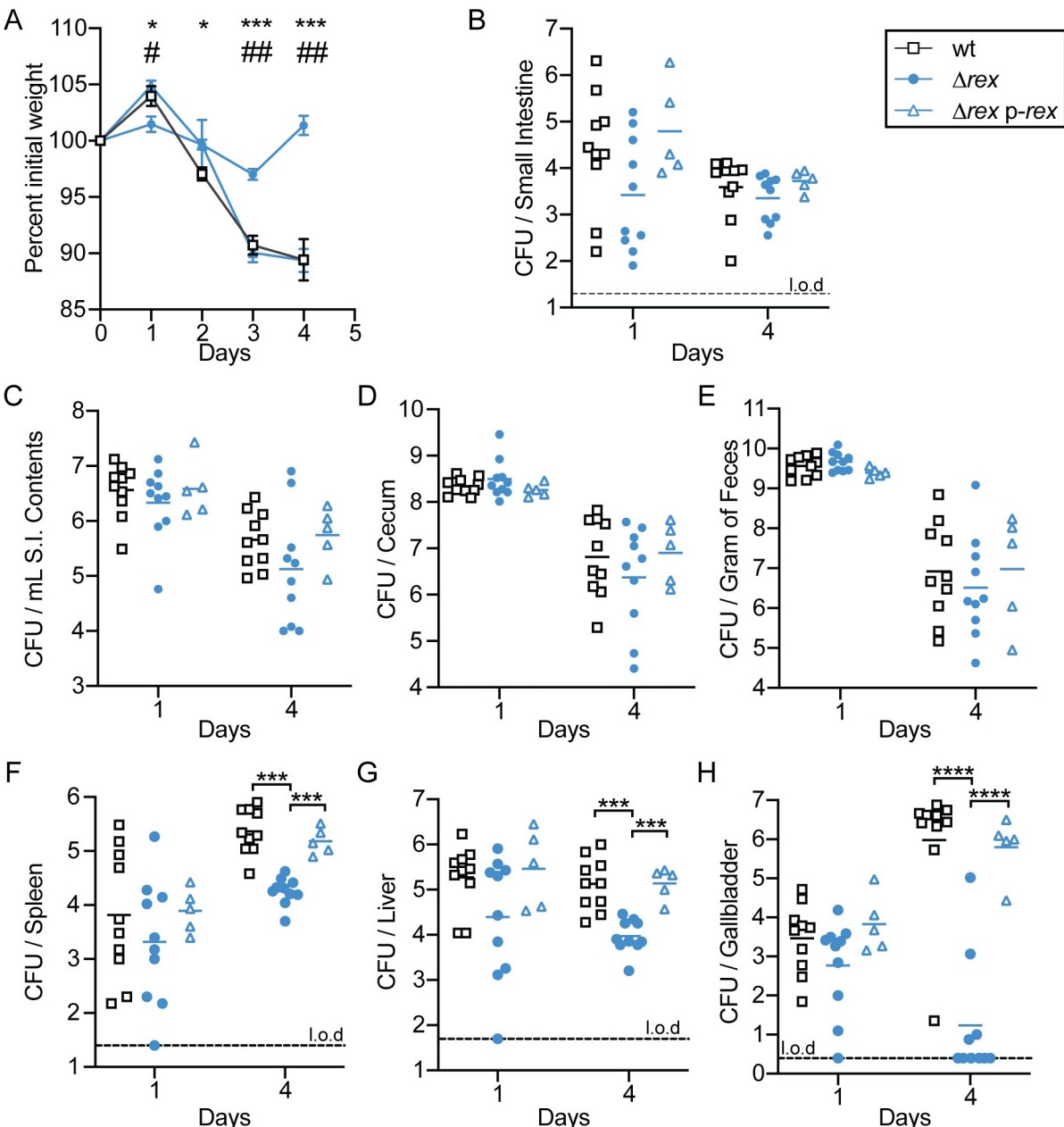

**Fig 5. Rex is required for full virulence in an oral listeriosis model.** Female BALB/c mice were orally infected with $10^8$ CFU of wt (black open squares), Δ*rex* (blue circles), or Δ*rex* p-*rex* (blue open triangles) and the number of bacteria present in each tissue was determined at 1 and 4 days post-infection. A. The body weights of the mice over time, reported as a percentage of body weight prior to infection. For wt and Δ*rex*, data are the means and SEMs of n = 30 (day 1), n = 20 (day 2), n = 15 (day 3), and n = 10 (day 4). For Δ*rex* p-*rex*, data represent the means and SEMs of n = 10 (day 1) and n = 5 (days 2–4). Significance between wt and Δ*rex* is denoted by *, while # indicates significance between Δ*rex* and Δ*rex* p-*rex*. B-H. Mice were sacrificed each day and organs were harvested to enumerate bacterial burden. Each symbol represents an individual mouse (n = 10 per group for wt and Δ*rex* and n = 5 per group for Δ*rex* p-*rex*) and the solid lines indicate the geometric means. Dashed lines indicate the limit of detection (l.o.d.). Data are combined from two independent experiments for the wt and Δ*rex* strains. Results are expressed as log-transformed CFU per organ or per gram of feces. *p* values were calculated using a heteroscedastic Student's *t* test. * or # $p < 0.05$; ## $p < 0.01$; *** $p < 0.001$; **** $p < 0.0001$.

Transcriptome analysis revealed that *L. monocytogenes* Rex regulates metabolic pathways similarly to what has been described in other Gram-positive bacteria [13, 15–17]. Specifically, fermentative metabolic pathways were the most significantly changed in the Δ*rex* mutant. We

identified putative Rex binding sites in the promoters of 48 Rex-repressed genes and/or operons, including those involved in fermentation and virulence, suggesting Rex likely binds and directly represses these genes. In contrast, genes activated by Rex lacked a Rex-binding site and we hypothesize they are indirectly regulated. Further protein-DNA binding analysis is needed to elucidate the direct regulon of *L. monocytogenes* Rex.

The results herein demonstrate that *L. monocytogenes* Rex functions to repress fermentation during aerobic growth in order to maximize energy generation. We found the Δ*rex* mutant over-expressed genes necessary for fermentative metabolism (*lap*, *ldhA*, and *pflBC/pflA*) and accordingly, produced more lactate and formate than wt when replicating aerobically. While acetate is the major by-product generated by wt *L. monocytogenes* during aerobic growth [29, 30], we observed a concomitant decrease in acetate production by Δ*rex*. Together, these results suggest that in the absence of Rex repression, there is an increased metabolic flux from pyruvate towards lactate and away from acetate production. The increased LdhA activity to produce lactate funnels NADH away from the ETC, resulting in less ATP generation by respiration. Indeed, the *L. monocytogenes* Δ*rex* strain exhibited an aerobic growth defect and produced half as much ATP as wt. The metabolic and growth phenotypes were ameliorated during anaerobic growth when Rex-mediated repression is relieved and fermentation is required for growth. Taken together, these results demonstrated that *L. monocytogenes* Rex is necessary to repress fermentative metabolism in the presence of oxygen in order to efficiently produce ATP.

Our transcriptional results suggested a role for *L. monocytogenes* Rex in regulating production of virulence factors necessary during oral listeriosis. Specifically, alleviation of Rex-mediated repression increased expression of genes encoding the bile detoxifying enzyme Bsh and the internalin proteins InlA and InlB. Within the GI tract, *L. monocytogenes* encounters acidified bile that can disrupt bacterial membranes, dissociate membrane proteins, and induce DNA damage and oxidative stress [9, 31]. Bsh detoxifies conjugated bile acids and contributes to bacterial survival in the GI tract, which is evidenced by the wide distribution of homologous enzymes among commensal gut bacteria [21, 22, 32]. Also within the GI tract, *L. monocytogenes* invades intestinal epithelial cells and disseminates to peripheral organs. InlA and InlB mediate invasion of non-phagocytic cells by engaging the host cell receptors E-cadherin and Met, respectively [5, 25, 33]. We demonstrated that *rex*-deficient *L. monocytogenes* were significantly more resistant to acidified bile stress and were better able to invade intestinal epithelial cells *in vitro*. It is important to note that while these *in vitro* experiments confirmed transcriptome analyses, it is unlikely that increased transcription of *inlAB* in the Δ*rex* mutant had any effect on this strain during the oral infection in mice. While InlAB-mediated invasion is required for invasion of non-phagocytes in cell culture, it has been reported that neither of these adhesins are required for successful oral infection in the mouse listeriosis model [28, 34, 35]. Together, our *in vitro* results suggest that alleviation of Rex-mediated repression coordinates expression of genes necessary in the GI tract during infection, including fermentative metabolism, bile resistance, and invasion of host cells.

Following invasion of intestinal epithelial cells, *L. monocytogenes* disseminates via the lymph and blood to the spleen and liver where it replicates intracellularly and spreads cell-to-cell without entering the extracellular space [4]. We found that Rex was dispensable for intracellular replication in activated bone marrow-derived macrophages and hepatocytes. Furthermore, *L. monocytogenes* Δ*rex* grew and spread cell-to-cell at the same rate as wt in both fibroblasts and hepatocytes. Combined, our *in vitro* results suggested that Rex is dispensable for the intracellular lifecycle and implied that Rex repression is alleviated in the GI tract during oral infection and this results in the upregulation of anaerobic metabolism and virulence factors.

To investigate the role of *L. monocytogenes* Rex in pathogenesis, we took advantage of a recently optimized oral listeriosis model of murine infection [8, 28]. As predicted from our *in vitro* results, we found that Rex was completely dispensable for colonizing the GI tract, suggesting that in wt *L. monocytogenes*, Rex repression is fully relieved in this hypoxic environment. Further, within the first 24 hours of infection, Δ*rex* disseminated to the spleen, liver, and gallbladder similarly to wt. However, the Δ*rex* mutant was attenuated overall, as mice infected with this strain lost significantly less body weight than mice infected with wt *L. monocytogenes*. Four days post-infection, we observed a 10-fold decrease in bacterial burden in the spleens and livers of mice infected with Δ*rex* compared to wt-infected mice. Surprisingly, Δ*rex* was attenuated ~5-logs in the gallbladders 4 days post-infection. This phenotype was quite dramatic, as only 2 of the 13 mice infected with Δ*rex* harbored more than 10 CFU in the gallbladders when all three infection experiments were combined. This was in stark contrast to mice infected with the wt or complemented strain, which each harbored approximately $10^6$ CFU per gallbladder at this time point.

A handful of other studies have identified *L. monocytogenes* mutants defective in colonizing the gallbladder, however these mutants were also more sensitive to bile stress *in vitro* or lacked known virulence factors [8, 23, 36–39]. In contrast, the Δ*rex* strain was insensitive to neutral bile, more resistant to acidified bile stress, and was not impaired for intracellular growth or intercellular spread. Other than bile stress, not much is known about impediments to bacterial proliferation in the gallbladder, despite its importance to *L. monocytogenes* pathogenesis. Early during infection, a few bacteria are released from lysed hepatocytes and transit through the common bile duct to colonize the lumen of the gallbladder where they replicate to very high densities [6–8]. After a meal, the gallbladder contracts and delivers bile along with a bolus of *L. monocytogenes* back into the small intestine where it can reseed the intestinal tract [40]. Thus, the gallbladder quickly becomes the primary reservoir of *L. monocytogenes* during infection [7]. Ongoing work is aimed at identifying the stressors present in the gallbladder that inhibit Δ*rex* replication in this organ, which may have more broad implications for other bacterial pathogens that replicate in the gallbladder, such as *Salmonella* spp [41].

Redox homeostasis and bacterial pathogenesis are intricately tied, although the mechanisms are not entirely understood [42]. Rex-regulated metabolic pathways have been indirectly implicated in virulence in other pathogens. *S. aureus* Rex controls expression of lactate dehydrogenase, which is essential for bacterial survival when exposed to nitric oxide produced by phagocytes [17, 43]. Similarly, *Clostridium difficile* Rex regulates butyrate production, which induces toxin synthesis during gut colonization [44]. In contrast, we identified putative Rex-binding sites in the promoters of *inlAB* and *bsh*, suggesting that *L. monocytogenes* Rex directly regulates these virulence factors. Interestingly, *bsh* and *inlAB* are positively regulated by the master virulence regulator PrfA and the stress-responsive alternative sigma factor SigB and are induced following exposure to bile and acidic conditions [21, 45–47]. This work demonstrated that *bsh* and *inlAB* are also induced under anaerobic conditions when Rex repression is alleviated. We predict these regulatory factors sense distinct or potentially overlapping environmental signals and converge on these virulence factors for appropriate and efficient regulation. Future research will investigate the crosstalk between these transcriptional regulators during pathogenesis and the variable redox environments encountered by *L. monocytogenes* during infection.

Overall, this work suggests a model in which *L. monocytogenes* Rex-mediated repression is alleviated in the anaerobic environment of the GI tract, thus upregulating fermentative metabolism and virulence factor production. However, following dissemination to internal organs, Rex is required to regulate factors critical for survival within the gallbladder. As the primary reservoir of *L. monocytogenes* during infection, identifying the factors required for survival

and replication in the gallbladder is imperative for understanding *L. monocytogenes* pathogenesis.

## Materials and methods

### Ethics statement

This study was carried out in strict accordance with the recommendations in the Guide for the Care and Use of Laboratory Animals of the National Institutes of Health. All protocols were reviewed and approved by the Animal Care and Use Committee at the University of Washington (Protocol 4410–01).

### Bacterial strains and culture conditions

*L. monocytogenes* mutants were derived from wild type strain 10403S [48, 49] and cultured in brain heart infusion (BHI) broth at 37˚C with shaking (220 rpm), unless otherwise stated. Antibiotics (purchased from Sigma Aldrich) were used at the following concentrations: streptomycin, 200 µg/mL; chloramphenicol, 10 µg/mL (*Escherichia coli*) and 7.5 µg/mL (*L. monocytogenes*); and carbenicillin, 100 µg/mL. Porcine bile (Sigma Aldrich) was dissolved in sterile BHI with streptomycin to ensure sterility. In cases where pH adjustments of media were carried out, 1N HCl was used and the pH was determined using VWR sympHony benchtop pH meter. *L. monocytogenes* strains are listed in Table F in S1 Text and *E. coli* strains are listed in Table G in S1 Text. Plasmids were introduced in *E. coli* via chemical competence and heat-shock and introduced into *L. monocytogenes* wt via trans-conjugation from *E. coli* SM10 [50].

### Cell lines

Huh7 and Caco-2 are cancer cell lines derived from human males with hepatocellular carcinoma and colon adenocarcinoma, respectively. TIB73 is a spontaneously immortalized hepatocyte cell line from a normal BALB/c embryo liver. Huh7, Caco-2, and TIB73 cell lines were obtained from Joshua Woodward (University of Washington) [51]. L2 fibroblasts were described previously [52]. Cell lines were grown at 37˚C in 5% $CO_2$ in Dulbecco's modified Eagle's medium (DMEM) with 10% heat-inactivated fetal bovine serum (FBS; 20% for Caco-2 cells) and supplemented with sodium pyruvate (2 mM) and L-glutamine (1 mM). For passaging, cells were maintained in Pen-Strep (100 U/ml) but were plated in antibiotic-free media for infections. Initial infection of TIB73 cells was carried out in DMEM with 0.1% FBS and replaced with 10% FBS in DMEM during gentamicin treatment.

### Vector construction and cloning

To construct the *Δrex* mutant, ~700 bp regions upstream and downstream of *rex* (LMRG_01223) were PCR amplified using *L. monocytogenes* 10403S genomic DNA as a template. PCR products were restriction digested and ligated into pKSV7-*oriT* [53]. The plasmid pKSV7xΔ*bsh* was constructed via Gibson assembly using the NEBuilder HiFi DNA assembly master mix. Regions ~1000 bp upstream and downstream of *bsh* were amplified with linker regions identical to those flanking a *ccdB* toxin cassette in pKSV7x [54]. pKSV7x was PCR amplified and DpnI treated. The linearized vector and insert PCR products were combined in the NEB master mix and ligated according to manufacturer instructions. pKSV7Δ*rex* and pKSV7xΔ*bsh* were transformed into *E. coli* and sequences were confirmed by Sanger DNA sequencing. Plasmids with the mutant Δ*rex* and Δ*bsh* alleles were introduced into *L. monocytogenes* via trans-conjugation and integrated into the chromosome. Colonies were purified on selective nutrient agar and subsequently cured of the plasmid by conventional methods [52].

Allelic exchange was confirmed by PCR. To construct the Δ*rex*Δ*bsh* mutant, pKSV7Δ*rex* was trans-conjugated into *L. monocytogenes* Δ*bsh* and integrated into the chromosome as described above.

To construct the Δ*rex*Δ*inlAB* mutant, the mutant *rex* region was amplified from pKSV7Δ*rex*, restriction digested, and ligated into the pLIM1 plasmid containing a PheS* counterselection marker (provided as a generous gift from Arne Rietsch, Case Western Reserve University). Sequences were confirmed by Sanger DNA sequencing. The plasmid was introduced into *L. monocytogenes* Δ*inlAB* via trans-conjugation and integrated into the chromosome as previously described [52, 55]. Briefly, transconjugants were selected by growing on BHI containing streptomycin and chloramphenicol at 30°C for 24 hours. A colony from this plate was re-streaked onto a fresh plate and incubated at 42°C for 24–48 hours. A colony was re-steaked and grown at 42°C two additional times to ensure integration of the pLIM1Δ*rex* plasmid into the chromosome. One colony was inoculated into BHI broth and grown overnight at 30°C. The culture was diluted $10^{-4}$ and 100 μl was plated on BHI agar supplemented with *p*-chloro-phenylalanine (18 mM) and incubated at 37°C overnight. Colonies that grew on the counterselection plates were validated to be chloramphenicol-sensitive and confirmed by PCR.

Ectopic expression of genes in *L. monocytogenes* was carried out using pPL2 integration plasmids [56]. The plasmid for complementing Δ*rex* was constructed by PCR amplifying *rex* along with its predicted native promoter using *L. monocytogenes* 10403S genomic DNA as a template. Sequences were confirmed by Sanger DNA sequencing. The constructed pPL2 plasmid was trans-conjugated into *L. monocytogenes* Δ*rex* and *L. monocytogenes* Δ*rex*Δ*inlAB* integration was confirmed by antibiotic resistance.

## RNA isolation

Nucleic acids were purified from bacteria harvested from broth culture as previously described [57]. Briefly, bacteria were grown overnight in BHI shaking at 37°C and normalized to an optical density at 600 nm ($OD_{600}$) of 0.02 into 25 mL BHI. After 4 and 7 hours of aerobic growth at 37°C shaking, bacteria were mixed 1:1 with ice-cold methanol, pelleted, and stored at -80°C. Bacteria were lysed in phenol:chloroform containing 1% SDS by bead beating with 0.1 mm diameter silica/zirconium beads. Nucleic acids were precipitated from the aqueous fraction overnight at -20°C in ethanol containing sodium acetate (150 mM, pH 5.2). Precipitated nucleic acids were washed with ethanol and treated with TURBO DNase per manufacturer's specification (Life Technologies Corporation).

## Transcriptomics

Ribosomal RNA was removed from total RNA samples using the Ribo-Zero rRNA Removal kit, according to manufacturer's recommendations (Illumina, Inc., San Diego, CA, USA). Depleted samples were analyzed and sequenced by the Genomics & Bioinformatics Shared Resources at Fred Hutchinson Cancer Research Center as previously described [58]. Results were evaluated using CLC Genomics Workbench (Qiagen) and transcripts that were changed >2-fold ($p < .01$) were included in our analysis. In addition, the data were technically validated by measuring expression of 6 genes via quantitative RT-PCR (qPCR) and a correlation was confirmed ($R^2 = .92$). In Table 1, we included genes of interest and those known to be regulated by Rex in other organisms.

## Quantitative RT-PCR of bacteria transcripts

Transcript analysis was performed as previously described [52]. Briefly, for the measurement of aerobic transcripts overnight cultures were normalized to an $OD_{600}$ of 0.02 in 25 mL BHI in

250 mL flasks and were incubated with shaking at 37˚C for 7 hours. For measurement of anaerobic transcripts, filter sterilized BHI broth was degassed overnight in an anaerobic chamber. Media was transferred to 16 x 125 mm Hungate Anaerobic Tubes (Chemglass Life Sciences) inside the anaerobic chamber and the tubes were autoclaved. The $OD_{600}$ was normalized to an $OD_{600}$ of 0.02 in 10 mL BHI in the Hungate tubes and incubated at 37˚C for 7 hours. Following 7 hours of both aerobic and anaerobic growth, RNA was harvested as described above with the exception of DNase treatment, which was carried out using Thermo Scientific DNaseI (Thermo Scientific) according to the manufacturer's instructions. The synthesis of cDNA was carried out using an iScript cDNA synthesis kit (Bio-Rad). Quantitative RT-PCR was performed on cDNA with the iTaq universal SYBR green supermix (Bio-Rad).

### Growth curves

For aerobic growth in broth, the cultures were normalized to an $OD_{600}$ of 0.02 in 25 mL BHI in 250 mL flasks and were incubated with shaking at 37˚C. The $OD_{600}$ was measured every hour. For anaerobic growth in broth, filter sterilized BHI broth was degassed overnight in an anaerobic chamber. Media was transferred to 16 x 125 mm Hungate Anaerobic Tubes (Chemglass Life Sciences) inside the anaerobic chamber and the tubes were autoclaved. The $OD_{600}$ was normalized to an $OD_{600}$ of 0.02 in 10 mL BHI in the Hungate tubes and incubated at 37˚C, with $OD_{600}$ measurements every hour.

### Measurement of bacterial metabolites

Bacteria from aerobic and anaerobic cultures were collected (1 mL aliquots) after 4 hours of growth and centrifuged at 13,000 x *g* for 2 min. The supernatants were removed, sterile filtered, and stored at -20˚C until use. Extracellular metabolites were determined using Roche Yellow Line Kits (R-Biopharm), according to the manufacturer's recommendation. Intracellular ATP concentrations were determined using a BacTiter-Glo kit (Promega) according to the manufacturer's protocol and normalized to optical units.

### Bile sensitivity assays

Overnight cultures were diluted 1:200 into BHI pH 5.5, BHI supplemented with 0.1% porcine bile, or BHI pH 5.5 supplemented with 0.1% porcine bile. Aerobic cultures were incubated for 24 hours at 37˚C shaking, followed by serial dilutions and plating on BHI agar to enumerate CFU.

### Bacterial invasion assays

Caco-2 or Huh7 cells were seeded $2.0 \times 10^5$ cells per well in 24-well plates and washed twice in sterile PBS just prior to infection. Bacterial cultures were incubated overnight in BHI broth at 30˚C static and then washed twice with sterile PBS and resuspended in cell culture media. Bacteria were added to cell monolayers at a multiplicity of infection (MOI) of 10 for Caco-2 cells and 20 for Huh7 cells. To measure bacterial invasion, monolayers were washed twice with PBS after 1 hour of infection and incubated with cell culture media containing gentamicin (50 μg/mL) for 1 hour. Monolayers were washed with PBS twice and lysed with 0.1% Triton X-100 and internalized bacteria were enumerated following plating on BHI agar [25, 59].

### Intracellular growth curves

Growth curves in bone marrow-derived macrophages (BMMs) were performed as previously described [58], with the following modifications. Briefly, BMMs were harvested as previously

reported [60] and seeded at a concentration of 6 x 10$^5$ cells per well in a 24-well plate the day before infection. BMMs were activated by incubating the monolayer with recombinant murine IFNg (100 ng/mL, PeproTech) overnight and during infection. Overnight bacterial cultures incubated at 30˚C statically were washed twice with PBS and resuspended in warmed BMM media [52]. BMMs were washed twice with PBS and infected at an MOI of 0.1. Thirty minutes post-infection, cells were washed twice with PBS and BMM media containing gentamicin (50 μg/mL) was added to each well. To measure bacterial growth, cells were lysed by addition of 250 μL cold PBS with 0.1% Triton X-100 and incubated for 5 min at room temperature, followed by serial dilutions and plating on BHI agar to enumerate CFU.

Growth curves in Huh7 and TIB73 cells were performed as previously described [51]. Briefly, Huh7 and TIB73 cells were seeded at a concentration of 2.0 x 10$^5$ cells per well in 24-well plates the day before infection. Overnight bacterial cultures incubated at 30˚C statically were washed twice with sterile PBS and resuspended in cell culture media. Huh7 and TIB73 cells were infected at an MOI of 20 or 50, respectively. Sixty minutes post-infection, cells were washed twice with PBS and cell culture media containing gentamicin (50 μg/mL) was added to each well. To measure bacterial growth, cells were lysed by addition of 250 μL cold PBS with 0.1% Triton X-100 and incubated for 5 min at room temperature, followed by serial dilutions and plating on BHI agar to enumerate CFU. Experiments were performed with technical replicates and repeated two times.

## Plaque assays

Plaque assays were performed as previously described [26, 52]. Briefly, 6-well plates were seeded with L2 fibroblasts or TIB73 cells at a density of 1.2 x 10$^6$ and 1.5 x 10$^6$, respectively. Bacterial cultures were incubated overnight at 30˚C in BHI broth and were then diluted in sterile PBS (1:10 for L2 infections; 1:2 for TIB73 infections). L2 fibroblasts and TIB73 cells were infected with 5 μL or 10 μL of diluted bacteria, respectively. 1 hour post-infection, cells were washed twice with sterile PBS and agarose overlays containing DMEM and gentamicin were added to the wells. 2 days post-infection, cells were stained with neutral red dye and incubated overnight. Plaques were imaged 72 hours post-infection and plaque area was quantified using Image J software [61].

## Mice

Female BALB/c mice were purchased from The Jackson Laboratory at 5 weeks of age and used in experiments when they were 6–7 weeks old. All mice were maintained under specific-pathogen-free conditions at the University of Washington South Lake Union animal facility. All protocols were reviewed and approved by the Animal Care and Use Committee at the University of Washington (Protocol 4410–01).

## Oral murine infection

Infections were performed as previously described [8, 28, 62–64]. Groups of 3 or 5 mice were placed in cages with wire flooring raised 1 inch to prevent coprophagy, and streptomycin (5 mg/mL) was added to drinking water 48 hours prior to infection. Food and water were removed 16 hours prior to infection to initiate overnight fasting. *L. monocytogenes* cultures were grown overnight in BHI broth at 30˚C static. The cultures were diluted 1:10 into fresh BHI broth and grown at 37˚C shaking for 2 hours. Bacteria were diluted in PBS and mice were fed 10$^8$ bacteria in 20 μL via pipette. The inocula were plated and enumerated after infection to ensure consistent dosage between strains. Food and water were returned to cages following infection and mice were euthanized at 1, 2, 3, and 4 days post-infection. Livers, spleens, and

feces were harvested and homogenized in 0.1% NP-40. Gallbladders were harvested and ruptured in 1 mL of 0.1% NP-40 with a sterile stick. The cecum sections were emptied, flushed with sterile PBS, and homogenized in 0.1% NP-40 buffer. The small intestines were cut lengthwise with sterile forceps and flushed with sterile PBS. Intestinal contents were resuspended in PBS and intestinal tissue was homogenized in 0.1% NP-40. All organs were serial diluted in PBS and plated on LB agar to enumerate CFU.

## Supporting information

**S1 Fig. Validation of transcriptome analysis by RT-qPCR.** Gene expression measured by quantitative RT-PCR following 7 hours of aerobic (A) and anaerobic (B) growth in the wt, Δ*rex*, and Δ*rex* p-*rex* strains. Data are graphed as the fold change over wt (wt = 1). In both panels, data are the means and SEMs of three independent experiments. Student's unpaired *t* test was used to compare fold changes between the Δ*rex* and wt strains and between the Δ*rex* and Δ*rex* p-*rex* strains (n.s., $p > 0.05$; *, $p < 0.05$; **, $p < 0.01$; ****, $p < 0.0001$). Data were not statistically significant between wt and Δ*rex* p-*rex*.
(TIF)

**S2 Fig. Growth and extracellular metabolite profiles are similar between the wt and Δ*rex* strains during anaerobic growth.** A. Anaerobic growth of wt and Δ*rex* strains, measured by $OD_{600}$. B-F. Supernatants were sampled at 4 hours during anaerobic growth. Concentrations of glucose (B), lactate (C), formate (D), acetate (E), and ethanol (F) were determined and normalized to $OD_{600}$. G. Concentration of glucose was measured in the supernatant over time. H. Concentration of ethanol in the supernatant over time, normalized to the $OD_{600}$ I. Relative intracellular ATP concentration was measured at 4 hours. In panels A-H, data are the means and SEMs of three independent experiments. Data in panel I is the mean and SEM of 2 independent experiments. A heteroscedastic Student's unpaired *t* test was used to compare results from wt and Δ*rex* (n.s., $p > 0.05$; ****, $p < 0.0001$).
(TIF)

**S3 Fig. Rex is dispensable for growth in acidified BHI.** Growth of wt (black), Δ*bsh* (grey), Δ*rex* (blue), and Δ*rex*Δ*bsh* (white) normalized to the initial inoculum (dashed line = 1). Strains were evaluated 24 hours post-inoculation in acidified BHI grown aerobically. Data are the means and range of three independent experiments. Strains were not significantly different (heteroscedastic Student's *t* test; $p > 0.05$).
(TIF)

**S4 Fig. Four day time-course of wt and Δ*rex* strains in an oral listeriosis model.** Female BALB/c mice were orally infected with $10^8$ CFU of wt (black squares) or Δ*rex* (blue circles) and the number of bacteria present in each tissue was determined over time. A. The body weights of the mice over time, reported as a percentage of body weight prior to infection. Data are the means and SEMs of n = 20 (day 1), n = 15 (day 2), n = 10 (day 3) and n = 5 (day 4). B-H. Mice were sacrificed each day and organs were harvested to enumerate bacterial burden. Each symbol represents an individual mouse (n = 5 per group), and the solid lines indicate the geometric means. Dashed lines indicate the limit of detection (l.o.d.). Results are expressed as log-transformed CFU per organ or per gram of feces. *p* values were calculated using a heteroscedastic Student's *t* test. * $p < 0.05$; ** $p < 0.01$; *** $p < 0.001$.
(TIF)

**S5 Fig. Wt and Δ*rex* strains in an oral listeriosis model 1, 2, and 4 days post-infection.** Female BALB/c mice were orally infected with $10^8$ CFU of wt (black squares) or Δ*rex* (blue

circles) and the number of bacteria present in each tissue was determined over time. A-F. Mice were sacrificed on 1, 2, and 4 days post-infection and organs were harvested to enumerate bacterial burden. Panel A includes small intestinal tissue only; bacterial burden in the intestinal contents was not evaluated. Each symbol represents an individual mouse (n = 3 per group) and the solid lines indicate the geometric means. Dashed lines indicate the limit of detection (l. o.d.). Results are expressed as log-transformed CFU per organ or per gram of feces. $p$ values were calculated using a heteroscedastic Student's $t$ test. $^*$ $p < 0.05$; $^{***}$ $p < 0.001$. (TIF)

**S1 Text. Table A in S1 Text.** All Rex repressed genes during stationary phase**. Table B in S1 Text** All Rex repressed genes during mid-log phase. **Table C in S1 Text** All transcripts less abundant in Δ*rex* during stationary phase. **Table D in S1 Text** All transcripts less abundant in Δ*rex* during mid-log phase. **Table E in S1 Text** Predicted Rex binding sites in the 10403S genome. **Table F in S1 Text** *L. monocytogenes* strains used in this study. **Table G in S1 Text** *E. coli* strains used in this study. (DOCX)

## Acknowledgments

The authors would like to thank Arne Rietsch (Case Western Reserve University) for the pLIM plasmid and Steve Libby (University of Washington) for technical assistance.

## Author Contributions

**Conceptualization:** Cortney R. Halsey, Michelle L. Reniere.

**Formal analysis:** Cortney R. Halsey.

**Funding acquisition:** Michelle L. Reniere.

**Investigation:** Cortney R. Halsey, Rochelle C. Glover, Maureen K. Thomason.

**Writing – original draft:** Cortney R. Halsey, Michelle L. Reniere.

**Writing – review & editing:** Cortney R. Halsey, Rochelle C. Glover, Maureen K. Thomason, Michelle L. Reniere.

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
