## [Decision Letter · Decision Letter 0]

22 Mar 2021

Dear Dr Reniere,

Thank you very much for submitting your manuscript "The global redox-responsive transcriptional regulator Rex represses fermentative metabolism and is required for Listeria monocytogenes pathogenesis" for consideration at PLOS Pathogens. As with all papers reviewed by the journal, your manuscript was reviewed by members of the editorial board and by independent reviewers. In light of the reviews (below this email), we would like to invite the resubmission of a significantly-revised version that takes into account the reviewers' comments.

We cannot make any decision about publication until we have seen the revised manuscript and your response to the reviewers' comments. Your revised manuscript is also likely to be sent to reviewers for further evaluation.

Sincerely,

Michael Wessels

Section Editor

PLOS Pathogens

Kasturi Haldar

Editor-in-Chief

PLOS Pathogens

orcid.org/0000-0001-5065-158X

Michael Malim

Editor-in-Chief

PLOS Pathogens

orcid.org/0000-0002-7699-2064

Reviewer's Responses to Questions

**Part I - Summary**

Reviewer #1: The manuscript by Halsey et al. reports on the role of the transcriptional repressor Rex in the regulation of Listeria monocytogenes (Lm) aerobic fermentative metabolism and virulence.

The authors first used RNA-Seq analysis to characterize Lm Rex regulon during aerobic exponential growth and in stationary phase. Repressed genes being supported by the presence of putative Rex binding sites in their promoters provided a list of Rex targets, encompassing on the one hand genes encoding proteins involved in fermentative metabolism, on the other hand, virulence genes. Metabolic analysis of mutant and complemented strains indicates that, by sensing the redox status, Rex represses fermentative pathways to favour aerobic respiration during aerobic growth, thus allowing faster growth rates. The authors also highlight a repressive action of Rex on bile salt hydrolase, compromising bacterial resistance to acid bile in aerobic conditions—but not in the anaerobic conditions encountered in the gut. They also show that Rex-mediated repression of inlAB is dampening Lm adherence to and invasion of host cells in vitro. Then, turning to oral infections of mice, the authors observed that the colonisation of deeper organs (liver, spleen and gallbladder) by Lm ∆rex was reduced compared to the wt, suggesting that while Rex function did not affect the survival and crossing of intestinal barrier in the anaerobic conditions of the gut, its repressive action on target genes was required at later stages of infection for efficient proliferation and survival in tissues.

This work provides novel insight into the coordination between redox sensing by Lm and its adaptation to an in vivo niche. This characterisation of Lm Rex regulon as well as its phenotype adds up to the complexity regulations that condition successful infection in vivo, and will be important to take into account in future works. A special effort was made to discriminate between the effects of Rex-mediated regulation on metabolism and on virulence

The study is overall convincing, well-thought and carefully performed, analysed and discussed in-depth. However, to strengthen conclusions, some experiments would require to be controlled with complemented strains, especially in vivo. A few minor inconsistencies in the presentation or interpretation of results also require being clarified. This is detailed in the comments below.

Reviewer #2: In this manuscript, Halsey et al investigate the role of the redox-sensitive transcriptional regulator, Rex, in gene expression and pathogenesis of Listeria monocytogenes. Using RNAseq, metabolite quantification, and in vitro and in vivo models of pathogenesis the authors identify genes that are potentially repressed by Rex and predict possible sites of direct binding proximal to the promoters of many of these genes, thereby implicating rex as a direct repressor. Given that Rex in other microrganisms is redox-responsive and senstive to shifts in aerobic or anaerobic metabolism, the authors confirm that Rex exerts control over the switch to fermentation of lactate and formate. They also determine that Rex appears to regulate expression of the bile salt hydrolase and internalins A nd B, linking rex to survival in the presence of bile, with additional potential impacts on Listeria invasion. Indeed, some cell lines, but not others display increases in adherence and internalization in a rex mutant and rex mutants are more resistant to porcine bile in vitro during aerobic growth at acidic pH. In vivo studies identify an important role for Rex in survival in the spleen and liver, but also more notably in the gall bladder where a rex mutant is dramatically attenuated. The major strengths of the study include the well-controlled and appropriate experiments and the intriguing in vivo defect in the gall bladder. The primary weaknesses of the study include a series of conclusions that do not completely align with the data provided, a lack of exploration as it relates to direct binding of rex to predicted promoter binding sites, conflicting data surrounding in vitro bile susceptibility phenotypes and in vivo phenotypes the gall bladder that aren't explored or discussed in much depth, and unclear significance of the internalin phenotypes.

Reviewer #3: The manuscript of Halsey et al., provides an initial characterization of Listeria monocytogenes Rex regulator, which is a conserved redox sensor/regulator in Gram positive bacteria. While the data suggsts that Rex represses fermentative metabolism and genes that play a role in Lm virulence, such as InlA/B and Bsh, to my opinion it is not strong enough and in some cases, it is even poorly presented (i.e., not statistically significant). It is therefore, many of the conclusions are not that convincing. I hope that the comments below will help to improve the manuscript for future submission.

**Part II – Major Issues: Key Experiments Required for Acceptance**

Reviewer #1: 1. Complementation of ∆rex strains.

Whereas the authors have generated a complemented ∆rex strain by introducing an ectopic version of rex in the genome, they only used these strains in some of their experiments. This could potentially mask side-effects of the initial deletion, either by a polar effect at the mutated locus, or in case another mutation occurred concomitantly with the deletion in the genome of that strain, on some of the assessed phenotypes.

It would thus be important to include the use of complemented strain in:

A. The in vivo experiments reported in Fig. 5;

B. The experiments which were carried-out to check by RT-qPCR the expression of six Rex targets as a validation of the transcriptome analysis (see also comment #2).

2. Validation of the transcriptome.

So that the reader can better assess the effect of Rex-mediated repression on its targets, the results of RT-qPCR experiments that were performed to validate the transcriptome should be included as a supplementary figure, together with data from a complemented strain. Moreover, because Rex seems to be exerting its repressive function during aerobic, but not anaerobic growth, it would be important to include both growth conditions in these graphs. Among the genes to be validated, if not those already assessed, it would be important to include the targets that the authors investigated in this work (especially bsh, inlAB, and at least 1or 2 of the metabolic genes). The assessment of rex expression in both aerobic and anaerobic conditions would also be informative, as well as expression in acid and neutral bile, aerobic and anaerobic conditions for rex and bsh (see comment #3).

Among the targets to be validated in the mutant and complemented strains for assessment of potential polar effects, beyond bsh (LMRG_01217) which is located within 5,000 base pairs of rex in the genome, LMRG_01221 and LMRG_01222, which are located immediately nearby rex and induced in ∆rex in stationary phase would also deserve testing.

3. Fig. 2. Role of Rex in bsh regulation and resistance to acid bile in anaerobic conditions.

During anaerobic growth in acid bile, the main difference that is seen with the ∆rex mutant strains came as non-significant in statistical testing. When looking at the data, it seems to proceed from a single data point behaving differently to the rest, which might as well be an experimental outlier. This would match the author’s statement that “Rex is typically derepressed during anaerobic growth”, hence wt and ∆rex are not expected to differ in this condition.

In addition, the interpretation provided in the text (P11 L188-200) when discussing the role of Rex on Bsh in the phenotype of the ∆rex∆bsh mutant is unconvincing, because in fact no effect of Bsh on survival in acid bile in anaerobic conditions is seen in this set of data, when comparing at the wt and ∆bsh strains. As stated by the authors, Rex is not active in anaerobic conditions; therefore in the wt strain bsh is expected to be expressed (unless it is repressed by something else in a Rex-independent way). In case Bsh was playing a role in acid bile resistance in this condition, viability of the ∆bsh strain should be lower than that of the wt strain (as was the case in aerobic conditions), which does not seem to be the case.

Then, the authors attribute, in the ∆rex strain, the increase in resistance to an increase in bsh expression, although (i) they did not assess this expression increase compared to the wt in anaerobic acid bile (see comment #2) and (ii) they assume that, due to the fact that Rex is not active in anaerobic growth, this increase in expression in the ∆rex mutant compared to the wt is “likely the result of activation by unknown factors” (i.e. Rex-independent, contradicting the statement L194 that “the increase in survival of ∆rex was dependent on bsh”).

Because the effect of ∆rex is not significant, because phenotypes are unconvincing, and because gene expressions underlying these phenotypes were not assessed, I suggest removing Fig. 2B and L189-197 from the results unless this experiment is consolidated. This would not be of any damage to the overall message of the article, since Rex is not expected to play a role in anaeroby. The last sentence of the paragraph would then read “Taken together, these results demonstrated that L. monocytogenes lacking the Rex repressor are more resistant to acidified bile due to increased bsh expression during aerobic growth”.

Reviewer #2: 1. The authors use in silico analysis to suggest that they have identified Rex binding sites in Listeria. Direct demonstration of binding to a representative site would affirm the notion that rex is a direct repressor.

2. Line 129 mentions genes "activated" by Rex. This does not seem to be an accurate statement as it implies Rex is a direct activator, which the authors assert is not likely to be true. There may be a better choice of words here.

3. Line 191-195 and Fig 2B. The statement that a rex mutant displayed a 20-fold increase in survival compared to WT is based off of a dataset where one data point biases the outcome, so it is difficult to make a conclusion one way or another. The possibility that a rex mutant cannot survive in 0.1% bile at pH5.5 when grown anaerobically is an equally viable conclusion, given the data. The experiment would have to be repeated with an increased number of replicates to make a conclusion on the increased survival of the rex mutant in this specific condition and the role of bsh in mediating that survival.

4. The only case where invasion appears to be impacted by a rex mutant is caco-2 cells. All other cell types (Huh-7, TIB73, and BMM) do not show any discernable differences at the 2-hour timepoint or at any timepoint thereafter (a consequence of increased invasion). Thus, it is unclear at this point whether or not the increased gene expression of inlA and inlB mediated by Rex manifests in any measurable way, or at all, during infection. The connection is not felt to be sufficiently justified based on the experimental evidence provided.

5. The dramatic decline in CFU in the gall bladder despite initially productive infection at day 1 is very cool, but is not congruent with the survival data for a rexA mutant in porcine bile. The animal data suggests that rex-mediated repression is quite important in the gall bladder despite its control over bsh. This makes it challenging to relate the in vitro findings to the in vivo condition. This is discussed by pointing out major unknowns about the gall bladder environment. Is there any way to more concretely reconcile these differences? Perhaps based on other gene expression changes that occurred?

6. Lines 278-280. It is stated that rex-dependent regulation is essential for surviving and colonizing the gall bladder. This statement is at odds with the data. The bacteria seem to get to the gall bladder just fine (infect?), and in some cases (not all) they either die off or are eliminated from the gall bladder by extrusion. Whatever the ultimate mechanism, two of the five mice still had a substantial number of CFU in the gall bladder, so the use of the term essential does not seem to be an accurate reflection of the data.

7. Lines 286-287. What is the evidence that Rex directly regulates virulence factor production during infection? The in vitro broth culture data suggest that it might occur, but there was no data to directly support regulation of virulence factors by Rex in vivo.

8. Lines 310-311 Is acetate fermentation really an "end-product" of aerobic growth? Acetate typically accumulates during overflow metabosim when there is excess carbon source available. The fact that it accumulates during aerobic growth is probably more a by-product of excess glucose in the medium, rather than it being an end-product of aerobic metabolism per se. Perhaps modify the language used?

Reviewer #3: Comments:

1. According to the literature, the regulation of Rex goes along with other stress regulators such as sigma B , and hence its examination under BHI conditions is probably masking for its true role. Personally, I think that these are not the best conditions to test the role of Rex (this is just a general comment, you don’t have to do with it anything).

2. It is necessary to validate the RNA seq data by RT-qPCR, at least for the genes that are investigated in the manuscript. The RT experiments should be done in aerobic and anaerobic conditions in WT versus delta-rex. It should be shown that the rex-dependent genes are indeed induced upon anaerobic growth in the presence of Rex, and even inhibited by Rex when over-expressed under this condition. These experiments will help to connect the identified genes with Rex-regulation under aerobic and anaerobic conditions.

3. Line 138-141: Figure 1B, the growth defect of delta-rex is minor and might be due to the high expression of its regulated genes, e.g., lap, which is highly induced (342-fold). The conclusion that Rex is dispensable under anaerobic growth is not supported by the presented data, as according to the growth curves, it seems to be dispensable also under aerobic conditions.

In this regard, it is strange that the authors could not detect ethanol production. What method was used?

4. Figure 1 D-G; a control is missing. The concentrations of these metabolites should be shown in WT bacteria grown in aerobic versus anaerobic conditions, to demonstrate the switch to fermentative metabolism, as well as in comparison to delta-rex, to examine its role in this regulatory switch.

5. Also in Fig 1C, a positive control for changes in glucose consumption is needed.

6. Fig 2A: the transcription level of bsh gene should be analyzed under the tested conditions, i.e., 0.1% bile and 0.1% bile+pH5.5 in WT and delta-rex mutants, to better connect the regulation of Bsh to the phenotypes.

7. Lines 192-197 (Fig 2B) should be rephrased, the phenotype of delta-rex is insignificant, and hence any interpretation of the data is not contributing.

8. It is worth over expressing Rex in WT bacteria grown under aerobic and anaerobic conditions with bile to see if the survival of the bacteria reduces.

9. Line 186-7, The data of the double mutant indicates that bsh deletion has a dominant effect. It will be nice to see if WT bacteria become resistant if over-expressing bsh (from pPL2).

10. Line 196- rex/bsh transcription during anaerobic growth should be measured by RT-PCR.

11. Fig 3 A-D. What are the p values for the differences between Δrex and Δrex pPL2rex? they seem to be not statistically significant. This could be problematic. Moreover, the phenotypes of the double mutant (with InlAB) could be due to the dominant effect of the internalins.

12. 229-231, the conclusion here is wrong. The data says that the rex-regulated genes are not required for intracellular growth, and not that the rex regulated genes are de-repressed during intracellular growth, that was not tested.

13. Can the authors show the intracellular growth curve in Caco-2, is there any defect?

14. Line 275-6. Rex repression can be tested experimentally. This is a major point in the model suggested by the authors, yet it is based on speculation and not on experimental data. In addition, ectopic expression of Rex may be helpful to decipher its importance in the in-vivo infection model.

15. Figure 5: this is the most important figure of the manuscript; however several issues arise:

• It is stated in the legend that solid line indicates the median of 5 mice per group. This representation is meaningless for n=5 and highly misleading. These should be presented as mean and error bars for standard deviation.

• I guess that panel A shows mean and stdev, but it is not stated in the legend.

• The authors should add the second biological repeat as a supplementary figure to give an estimation of the biological reproducibility.

• The experiments should include the complemented strain Δrex-pPL2-rex.

• Fig 5B shows a 3-log difference at day 1, while the other panels do not. Can the authors comment on the biological relevance of that, or that this is a technical problem?

• Since the authors suggest that Rex is not required in the GI, similar results for dissemination to organs are expected also upon IV infection. Did the authors perform these experiments?

• Lastly, the suggested role of Rex is not in line with the in vivo data (this mutant is attenuated in mice). As well as the bile data is not in accordance with the survival of the bacteria in the gallbladder. Can the authors comment on that?

**Part III – Minor Issues: Editorial and Data Presentation Modifications**

Reviewer #1: 4. Fig. 3. Role of Rex in inlAB regulation and effect in cell adherence/internalisation.

According to the description in the methods section (p. 23), the ∆rex∆inlAB strain that was used in Fig. 3 was not obtained by deleting inlAB in the ∆rex strain that had been previously generated, and that was checked by pPL2rex complementation on the same graph, but by deleting rex in a ∆inlAB strain previously generated in another lab. In addition, another vector (pLIM∆rex) was used for generating this construct than the pKSV7∆rex construct used when generating the ∆rex strain. Altogether, this makes it difficult to assess whether the ∆rex∆inlAB strain might not suffer from other mutations introduced during rex mutagenesis than rex deletion itself.

5. Fig. 5. Phenotype in the gallbladder.

In the abstract, the authors make a point that the ∆rex mutant was “nearly sterilized in the gallbladder”. This notion of a phenotype that would be more pronounced in the gallbladder than in other deeper organs analysed (spleen, liver) is also emphasised in the discussion section, although not strongly supported by the data provided, given (i) the high variability in counts in this organ and (ii) the fact that the mutant was not complemented (see comment #1).

This insistence on a phenotype in the gallbladder may also bring confusion to the reader, due to the repressive role of rex on bsh and the effect of the ∆rex mutant in acid bile, by a juxtaposition effect when flipping through the abstract without background. In the current state of the data, being less affirmative about a role in the gallbladder might be cautious.

Related with that, P19 L365, “specifically” may be too strong, as the authors do not really show that ∆rex is more “specifically” affected in the gallbladder than elsewhere. Lower counts on average plus high variability in this organ could also proceed from bottleneck effects on the small counts of bacteria being liberated from the liver.

6. Mouse model.

The authors used a conventional mouse model, rendered more susceptible to listeriosis thanks to antibiotic treatment prior to gavage. However, conventional mice are non-permissive to InlA-mediated entry in epithelial cells, due to an amino-acid change in the E-cadherin receptor (Lecuit et al. EMBO J 1999). Because the authors show that Rex is a regulator of inlA expression, they should at least mention in the discussion that, in case Rex regulated InlA-mediated invasion in vivo, the chosen mouse model would have been blind to any possible consequence on infection. Especially, possible effects on the ability to cross the intestinal barrier or to colonize tissues may have been missed.

7. It would be informative to indicate the position of rex in the genome (LMRG_01223), for instance in the methods section, in the paragraph dedicated to the description of how mutant strains were generated.

8. Use of “derepression”.

Throughout the text, the used or “Rex derepression” by the authors may be confusing to the reader. Indeed, while the authors mean that Rex target genes are derepressed (that the function of the repressor is alleviated), the reader might interpret this ambiguous wording as the alleviation of a repression that was exerted on rex function (meaning that the repressor would become active). Here follow a few suggestions of rewording to avoid this ambiguity:

P10 L160-1. “demonstrating that Rex-mediated repression is normally alleviated in this growth environment”

P11 L196. “As Rex repressor activity is typically turned off during anaerobic growth”

P13 L229. “These results suggest that Rex repressive activity on its genes is alleviated in wt L. monocytogenes during intracellular growth”

P13 L243. “Rex-mediated repression”

P13 L255. “Derepression of Rex targets would not”

P16 L285. “that derepression of Rex targets”

P17 L318. “when Rex-mediated repression is relieved”

P17 L323-4. “Alleviation of Rex-mediated repression increased expression”

P18 L 335. “showed that alleviation of Rex-mediated repression coordinates”

9. P19 L356, the statement “although bile was not toxic to L. monocytogenes at neutral pH” is misleading in this position in the text, because it seems to imply that Rex could have been expected to participate in regulating the detoxification against bile (actually, to help resist against bile, since here ∆rex is attenuated). However, the authors showed previously that Rex did not participate in resistance against neutral bile in vitro (Fig. 2A). It is thus useless, and even confusing to readers to relate again Rex to bile only because the gallbladder is mentioned, rather than stating, as for the liver or spleen, that Rex is required for proliferation and maintenance in this organ.

10. Statistics.

In most graphs, statistical testing was performed by heteroscedastic Student’s t-test. Because the data tested are in most figures not from two independent groups, but from three or more (for instance in Fig. 2, 5 conditions are tested and compared with each other), one-way ANOVA followed by appropriate post-hoc test would be better suited.

Reviewer #2: (No Response)

Reviewer #3: Comments about the bioinformatic data:

Line 120-121:"We identified potential Rex binding sites in the promoter regions of 48 genes and/or operons repressed by Rex (Table S5)." How this analysis was conducted? What was the genome analyzed? 10403S? If so the gene tags should be LMRG_XXXX and not LMO.

Tables S1 and S2, column titles (LMRG and Lmo) should be replaced with names of corresponding strains - 10403 and EGD-e.

A comprehensive info of the Supplementary Tables (S1, S2, and S5) should be updated in terms of ‘annotation’ of function(s) on many “hypothetical proteins” shown there. The current NCBI and UniProtKB resources can help to do that. For example the following “hypothetical proteins” have an annotation: LMRG_02136 is CRISPR/Cas system-associated protein Cas2 (cl11442); lmo2234 is predicted to be a sugar phosphate isomerase/epimerase (COG1082) and it has an excellent resolution 3D structure (1.7A; PDB: 2G0W); lmo2237, is a putative member of the well-known Major Facilitator Superfamily (MFS) protein (cl28910) etc. This information will give more insight into the impact of Rex.

The lap gene (LMRG_01332, lmo1634) found in this study as Rex-dependent (342-fold; Tables 1 and S1) is probably one of most abundant in expression even in WT bacteria. Wurtzel et al (2012) showed its hyperexpression in str. EGD-e grown in BHI (see data in their Table S1). Also, the two predicted 3-mismatch Rex ROPs of lap (CACGTGAAACACTGGACAAA, TTTGTGAAGTTTTTCACGTG) should be replaced to be shown according to their chromosome positions (TTTGTGAAGTTTTTCACGTG, CACGTGAAACACTGGACAAA); these two candidates overlap one another – any comments on that!? See a similar story of double and/or overlapping binding sites in CodY regulons of both L. monocytogenes and B. subtilis. Other ‘double’ ROPs in Table S1 should be verified e.g. of lmo1945.

The data presented in Table S5 should be used to get a preliminary consensus of Rex by using WebLogo options (http://weblogo.berkeley.edu/logo.cgi). I used the whole set of 0-3 mismatch predicted ROPs of Table S5 and found that there are some differences between the L. monocytogenes Rex ROP consensus and those of B. subtilis and S. aureus. There is a trend to have W-rich (A or T nucleotides) linker between left and right arms of the L. monocytogenes ROP consensus. Prediction of the ‘listerial’ consensus of Rex ROP could make the analysis of putative members of the regulon more intriguing.

Hecker and his colleagues wrote in 2009 (Res Microbiol) - "The...question is: Why do all Rex regulated genes not behave in the same way? Obviously, there is fine adjustment in expression of many anaerobically induced genes that need, in addition to inactivation of Rex, a second regulatory protein that activates their transcription under anaerobic conditions."

Minor comments:

1. misspelling in Fig S1 line 161

2. Correct fonts in figure legend Fig 2S

PLOS authors have the option to publish the peer review history of their article (what does this mean?). If published, this will include your full peer review and any attached files.

Reviewer #1: **Yes: **Alice Lebreton

Reviewer #2: No

Reviewer #3: No
---

## [Decision Letter · Decision Letter 1]

2 Jul 2021

Dear Dr Reniere,

Thank you very much for submitting your manuscript "The redox-responsive transcriptional regulator Rex represses fermentative metabolism and is required for Listeria monocytogenes pathogenesis" for consideration at PLOS Pathogens. As with all papers reviewed by the journal, your manuscript was reviewed by members of the editorial board and by several independent reviewers. The reviewers appreciated the attention to an important topic. Based on the reviews, we are likely to accept this manuscript for publication, providing that you modify the manuscript according to the review recommendations.  

Specifically there are a few points made by Reviewer 1 and Reviewer 4 that could benefit from clarification/modification in the text.   While Reviewer 4 does suggest additional experiments that may be useful for you to read, the focus of the minor revisions should be to clarify or modify your interpretation of the data taking into account the reviewer comments.

Sincerely,

Mary O'Riordan

Associate Editor

PLOS Pathogens

Michael Wessels

Section Editor

PLOS Pathogens

Kasturi Haldar

Editor-in-Chief

PLOS Pathogens

orcid.org/0000-0001-5065-158X

Michael Malim

Editor-in-Chief

PLOS Pathogens

orcid.org/0000-0002-7699-2064

Reviewer Comments (if any, and for reference):

Reviewer's Responses to Questions

**Part I - Summary**

Reviewer #1: In the revised version of their manuscript, Halsey et al. have addressed my comments on the previous version, and either provided new results that substantiate their findings, or discussed their claims appropriately with regards to the state-of-the art. In the few instances where the authors claims were not totally substantiated in the first version, careful rephrasing of the message was conducted, which toned it down. The authors also brought clarifications that improve reading in the discussion section. I commend the quality of the newly added data that now makes the authors’ demonstration compelling.

Based on these improvements, as well as — as far as I can judge — the replies provided to the other reviewers' concerns, I would recommend its publication.

Reviewer #2: The authors have addressed my concerns with well-reasoned arguments and appropriate modifications to the manuscript. I have no addtional concerns.

Reviewer #4: The manuscript by Halsey et al. investigates the function of Rex in Listeria monocytogenes, with an emphasis on infection and GI tract-relevant conditions. The authors identified Rex-repressed genes during growth in rich BHI broth and validated several genes for expression in the delta-rex mutant. Among these genes, bsh and inlAB are especially novel and interesting. The authors then showed that, consistent with Rex-mediated repression of these genes, the delta-rex mutant was enhanced for bile stress survival and non-phagocytic cell invasion. However, these in vitro phenotypes did not relate with oral mouse infection data. Interestingly, by contrast to in vitro phenotypes, the delta-rex mutant was attenuated in systemic organs and the gall bladder.

Overall, the manuscript presents several useful datasets such as the identification of Rex regulon in Listeria, and the most novel aspect is perhaps the characterization of Rex during infection. However, as they are presented, the data are mostly observational. The major question arises as for the discrepancy between in vitro data (bile resistance and epithelial cell invasion) and in vivo phenotypes (no role for Rex in GI tract and its requirement for Listeria replication in systemic organ, but only following oral infection). I have a few suggestions to strengthen the conclusions and increase the impact of the manuscript as presented below.

**Part II – Major Issues: Key Experiments Required for Acceptance**

Reviewer #1: (No Response)

Reviewer #2: N/A

Reviewer #4: 1. Aerobic/fermentative metabolism by the rex mutant

- The aerobic growth defect of the rex mutant, although statistically significant, is very small. BHI likely has excessive fermentable substrates, allowing both fermentation and oxidative phosphorylation to occur. I suggest that the authors revisit aerobic growth in a defined medium, as they also briefly mentioned in the discussion. This is an easy experiment that will substantially strengthen the conclusion.

- In Fig. 1G: ATP production by WT and delta-rex in the absence of oxygen should be presented as relative to WT grown aerobically. If increased fermentation reduces ATP synthesis, then anaerobic WT should have less ATP than aerobic WT.

2. Bile survival phenotype

- The bile exposure experiments were performed for 24 hours. This seems like a very long time, in comparison to 8 hours of exposure described in Dowd et al. 2011. I am concerned that such a long exposure may produce artifactual data due to spontaneous suppressors and altered metabolism in prolonged stationary phase.

Additionally, data in Figure 2 do not prove that increased bile resistance in the delta-rex mutant is due to increased bsh expression, since this was not directly quantified under the tested condition. I have read the authors’ explanation that they could not obtain quality RNA from bile-treated cultures, and that bsh over-expression was technically difficult. As an alternative approach to evaluate the bsh regulation by Rex, why not test the delta-rex delta-bsh pPL2-bsh strain (driven by a chemically-inducible promoter), which is not subject to Rex regulation? If performed, the experiment will strengthen the conclusion and show a physiological relevance for bsh regulation by Rex.

3. Tissue culture infection

- In Fig. 3, what are the Caco-2 and Huh7 infection rates by WT? I recall from the literature that 10403S invades Caco-2 cells at ~1%. If so, the small increase in invasion by the delta-rex mutant may not have much biological significance.

- Similar to the bile survival data, Figure 3 does not prove that increased Caco-2 and Huh7 invasion by delta-rex is due to increased inlAB expression, which was not quantified for Listeria cultures grown for infection – a different condition than that for RNASeq and qPCR experiments. Furthermore, at 2 hours post infection, Listeria would have gotten out of the phagosome, so the data here reflect both endocytosis and vacuolar survival/escape. Finally, growth curves in Figure 4 and mouse infection data in Figure 5 suggest that the small difference exhibited by delta-rex in early time points does not impact infection kinetics or outcomes. I’m not asking the authors to do more experiments here, but I’d suggest that they refrain from attributing the small phenotype in Figure 3 to inlAB expression, and discuss the physiological relevance of this data.

- Because the increase in Caco2 and Huh7 initial infection did not translate to GI tract infection, the discussion in lines 344-347 and 353-356 seems moot and misleading. I suggest this to be removed.

4. Mouse infection studies

- Data in figure 5 are the most substantial and novel aspect of the manuscript. I think that the lack of delta-rex phenotype in GI tract infection is consistent with the microaerophilic/anaerobic condition in the GI tract, under which Rex does not repress target genes.

- I find data in Fig 5F-G-H and Fig. S4 very intriguing, as they suggest that the delta-rex mutant is defective for replication in systemic organs, even though it initially reaches these organs equally well compared to WT. I noticed that the authors have performed a preliminary intravenous infection for WT and delta-rex, and found them to exhibit similar burdens in the spleen and liver at 48hpi. This data is consistent with Fig. S4 in which delta-rex defect only occurs between 2 and 4 dpi. So I strongly encourage the authors to repeat intravenous infection for more replicates and assess burdens at 2 and 4dpi. Validating the importance of Rex during systemic infection would substantially increase the impact of the manuscript.

**Part III – Minor Issues: Editorial and Data Presentation Modifications**

Reviewer #1: In the discussion, I would suggest rephrasing the following sentence in the final version:

P17L342-344 “While InlAB-mediated invasion is required for invasion of non-phagocytes in cell culture, it has been reported that neither of these adhesins are required for successful infection within a mouse” might be amended to “While InlAB-mediated invasion is required for invasion of non-phagocytes in cell culture, it has been reported that neither of these adhesins are required for successful intestinal infection in mice.” (or "oral infection in the mouse listeriosis model").

Indeed, while mouse E-cadherin is insensitive to InlA, mouse Met is a target of InlB. Only the absence on the InlB-mediated pathway in the gut makes mice insensitive to both InlA and B during oral infection. In contrast, the intravenous route is InlB-sensitive in mice.

Reviewer #2: N/A

Reviewer #4: None

PLOS authors have the option to publish the peer review history of their article (what does this mean?). If published, this will include your full peer review and any attached files.

Reviewer #1: **Yes: **Alice Lebreton

Reviewer #2: No

Reviewer #4: No

Figure Files:

Data Requirements:

Reproducibility:

References:

---

## [Editor Report · Decision Letter 2]

27 Jul 2021

Dear Dr Reniere,

We are pleased to inform you that your manuscript 'The redox-responsive transcriptional regulator Rex represses fermentative metabolism and is required for Listeria monocytogenes pathogenesis' has been provisionally accepted for publication in PLOS Pathogens.

Best regards,

Mary O'Riordan

Associate Editor

PLOS Pathogens

Michael Wessels

Section Editor

PLOS Pathogens

Kasturi Haldar

Editor-in-Chief

PLOS Pathogens

orcid.org/0000-0001-5065-158X

Michael Malim

Editor-in-Chief

PLOS Pathogens

orcid.org/0000-0002-7699-2064
---

## [Editor Report · Acceptance letter]

10 Aug 2021

Dear Dr Reniere,

We are delighted to inform you that your manuscript, "The redox-responsive transcriptional regulator Rex represses fermentative metabolism and is required for Listeria monocytogenes pathogenesis," has been formally accepted for publication in PLOS Pathogens.

Best regards,

Kasturi Haldar

Editor-in-Chief

PLOS Pathogens

orcid.org/0000-0001-5065-158X

Michael Malim

Editor-in-Chief

PLOS Pathogens

orcid.org/0000-0002-7699-2064